# A comparison of attitudes and knowledge of pre-exposure prophylaxis (PrEP) between hospital and Key Population Led Health Service providers: Lessons for Thailand's Universal Health Coverage implementation

Ajaree Rayanakorn[1,2], Sineenart Chautrakarn[1], Kannikar Intawong[1], Chonlisa Chariyalertsak[1], Porntip Khemngern[3], Debra Olson[4], Suwat Chariyalertsak[1]*

1 Faculty of Public Health, Chiang Mai University, Chiang Mai, Thailand, 2 School of Pharmacy, Monash University Malaysia, Jalan Lagoon Selatan, Bandar Sunway, Selangor, Malaysia, 3 Division of AIDS and STIs, Department of Disease Control, Ministry of Public Health, Nonthaburi, Thailand, 4 Professor Emeritus, School of Public Health, University of Minnesota, Minneapolis, Minnesota, United States of America

* suwat.c@cmu.ac.th

## Abstract

### Background

HIV Pre-exposure prophylaxis (PrEP) has demonstrated efficacy and effectiveness among high-risk populations. In Thailand, PrEP has been included in the National Guidelines on HIV/AIDS Treatment and Prevention since 2014. As a part of the national monitoring and evaluation framework for Thailand's universal coverage inclusion, this cross-sectional survey was conducted to assess knowledge of, attitudes to and practice (KAP) of PrEP service providers in Thailand.

### Methods

We conducted a cross-sectional survey to explore knowledge of, and attitudes towards PrEP among providers from hospital and Key Population Led Health Services (KPLHS) settings. The questionnaire was distributed online in July 2020. Descriptive and univariate analysis using an independent-sample t-test were applied in the analyses. Attitudes were ranked from the most negative (score of 1) to the most positive (score of 5).

### Results

Overall, there were 196 respondents (158 from hospitals and 38 from KPLHS) in which most hospital providers are female nurse practitioners while half of those from KPLHS report current gender as gay. Most respondents report a high level of PrEP knowledge and support provision in all high-risk groups with residual concern regarding anti-retroviral drugs resistance. Over two-fifths of providers from both settings perceive that PrEP would result in risk compensation and half of KPLHS providers are concerned regarding risk of sexual transmitted infections. Limited PrEP counselling time is a challenge for hospital providers.

**Data Availability Statement:** All relevant data are within the paper and its Supporting Information files.

**Funding:** The work is supported by The Global Fund to Fight AIDS, Tuberculosis and Malaria (GFATM), Department of Disease Control, Ministry of Public Health (Grant No. 7/2563) and The Joint United Nations Programme on HIV/AIDS (Grant No. 2020/1025705). The funders had no role in study design, data collection and analysis, decision to publish, or preparation of the manuscript.

## Conclusions

Service integration between both settings, more involvement and distribution of KPLHS in reaching key populations would be essential in optimizing PrEP uptake and retention. Continuing support particularly in raising awareness about PrEP among healthcare providers and key populations, facilities and manpower, unlimited quota of patient recruitment and PrEP training to strengthen providers' confidence and knowledge would be essential for successful PrEP implementation.

## Introduction

Thailand is among the countries with the highest HIV prevalence in Asia and the Pacific region with an estimated 480,000 people living with HIV in 2018 accounting for 9% of the region's total HIV infected population [1]. Even though the prevalence has been declining over the past decades due to successful early treatment and prevention programs, HIV remains a significant public health issue of the country. To achieve the country's ambitious goal in stopping the AIDS epidemic by 2030, a number of strategies including scale-up screening for early HIV treatment and HIV prevention are required to pave the way for disease eradication.

HIV Pre-exposure prophylaxis (PrEP) is the use of antiretroviral medications among uninfected individuals at high risk for HIV infection. The combination of two antiretroviral drugs, 200 mg emtricitabine (FTC) and 300 mg tenofovir disoproxil fumarate (TDF), has been recommended by the World Health Organization (WHO) for use as PrEP in HIV/AIDS before potential HIV risk exposure [2]. The use of medications consistently and correctly has been clinically proven to be effective, and cost-effective among populations with substantial risk of HIV infection [3–6]. In Thailand, PrEP has been included in the National Guidelines on HIV/AIDS Treatment and Prevention by the Ministry of Public Health (MoPH) as an additional measure among high-risk populations since 2014 [7] and become available free of charge under the Universal Health Coverage (UHC) since 2019 [8].

PrEP service in Thailand has been run as a pilot program supported by different sources of funding from both public and private sectors including the Joint United Nations Programme on HIV/AIDS, the Thai Red Cross Princess Soamsawali HIV Prevention (known as 'Princess PrEP Program') and the Global Fund to Fight AIDS, Tuberculosis and Malaria. More than 23,000 individuals have been served with PrEP to date [9]. The service can be provided either at hospitals or managed by a network of various community-based organizations where PrEP clients can go to community-based Key Population Led Health Services (KPLHS) for medications and follow-up. The KPLHS model was established in 2015 in response to the needs of the key populations [10,11] at risk for HIV. A defined set of HIV-related health services is provided at KPLHS clinics which are in close proximity to key population communities [12]. The main source of fundings for KPLHS is largely from non-government organizations. The first KPLHS program for marginal population was supported by the Princess Soamsawali HIV Prevention fund under the Thai Red Cross AIDS Research Center of which PrEP services were provided in eight clinics in four provinces (Bangkok, Chonburi, Chiang Mai and Songkhla) [13]. In contrast to the hospital-based model where PrEP is delivered at hospitals by healthcare practitioners, at KPLHS, PrEP is provided by lay providers who are often members of the key populations under a "needs-based, demand-driven, and client-centred" approach in close collaboration with the public health sector to ensure friendly and respectful service access [10,14]. The model has demonstrated feasibility and accessibility in reaching high-risk individuals who

are men who have sex with men (MSM), transgender women (TGW), and sex workers (SWs), and people who inject drugs (PWID) contributing two-thirds of new HIV cases in Thailand [15]. The cumulative number of PrEP users in Thailand has been increasing substantially from 1,865 in 2017 to 13,769 in 2021, of which KPLHS accounted approximately two-third of PrEP services [16].

To ensure sustainable service delivery, the National Health Security Office (NHSO) launched a pilot project in 2020 to provide PrEP for 2,000 new clients at 50 PrEP service centers (46 hospitals and 4 KPLHS) in 21 provinces across the country. In this regard, the national monitoring and evaluation (M&E) framework has been adopted to evaluate early implementation. As a part of the M&E process, this cross-sectional survey was conducted to explore knowledge of, attitudes to and practice of PrEP service providers from both hospital and KPLHS settings. The insights gathered would be useful for planning the national PrEP program roll-out under the country's UHC to maximize its uptake and retention with the ultimate goal of ending AIDS by 2030.

## Methods

### Design

An anonymous cross-sectional online survey was conducted to examine PrEP service providers' knowledge of and attitudes towards PrEP service. The survey is comprised of three parts. The first part captures demographic information including age, sex by birth (male, female) and current gender/sexual orientation (gay, bisexual, TGW, other), profession, years of experiences in PrEP services, and their self-knowledge rating about PrEP service from 0 (very poor knowledge) to 9 (very good knowledge). The second part entails attitudes towards PrEP service in terms of the evidence base, experiences, prioritization, effectiveness among risk groups, required support from NHSO/MoPH, and provision. Multi-item scales ranging from 1 to 5 were applied for all attitude questions ranking from the most negative (score of 1) to the most positive attitudes (score of 5). The third part inquires about the first three supports required from NHSO/MoPH regarding PrEP service in comments based on their opinion. The study partially applied similar domains to those used in a survey among Italian healthcare practitioners (HCPs) by Puro et al. [17] and in a survey of UK HCPs by Desai et al. [18]. The final survey used in this study was developed by the M&E research team with inputs from the national PrEP working committee under the Department of Disease Control, Ministry of Public Health. The study was approved by the Research Ethics Committee, Faculty of Public Health, Chiang Mai University, Thailand (Document No. ET017/2020). Respondents were informed that their participation was voluntary and willingness to participate was confirmed and checked at the beginning of the online questionnaire in order to proceed with the completion.

### Participant recruitment

A Quick Response (QR) code, a type of barcode containing information as "a series of pixels in a square-shaped grid" which can be read easily by a digital device [19] was used to store the anonymous self-administered online survey and distributed to all 50 active PrEP centers (46 hospitals and 4 KPLHS) across the country in 21 provinces. In addition, the online survey link and QR code were also provided via a social media chat group under LINE, the most-downloaded social media application for instant messaging [20] among PrEP service providers under NHSO and the national PrEP working committee created as an internal communication channel for monitoring and consultation. The participants were staff engaging in PrEP service delivery and counselling to PrEP clients. The survey responses were collected in July 2020.

## Data analysis

STATA version 16.1 (College Station, Texas, USA) was used for statistical analysis. Descriptive analysis was applied for demographic data overall and by type of service delivery model (KPLHS and hospital). Attitudes were ranked from the most negative to the most positive among all respondents in which the responses were categorized into negative attitude (strongly disagree and disagree), undecided, and positive attitude (agree and strongly agree) using positive statements for ease of comparison (the original statements were provided in S1 Table). Univariate analysis using an independent-sample t-test was employed to determine the difference of the mean scores of self-rated PrEP knowledge between KPLHS and hospital groups. Free text responses were quantified and coded into similar categories. Missing values were excluded from analyses.

## Results

### Respondent characteristics

The survey was completed by 196 PrEP service providers from all 50 PrEP service centers (46 hospitals and 4 KPLHS) under NHSO in Thailand. The four KPLHS were Rainbow Sky Association of Thailand (RSAT), Service Workers in Group (SWING), MPlus, and Caremat. MPlus delivers PrEP service on behalf of Nakornping hospital while PrEP service at SWING, RSAT and Caremat was operated on behalf of the Thai Red Cross AIDS Research Center (Anonymous Clinic). Among these, 38 people were from 4 KPLHS while 158 respondents were from 46 hospitals. The majority of participants sex by birth from hospitals were female (n = 132, 84.54%) whereas most service providers from KPLHS were male (n = 34, 89.47%). Half of these male service providers by current gender were gay, and nearly one fourth (n = 9, 23.68%) were TGW, while one each identified themselves as 'bisexual', and other. The mean age of respondents from hospitals and KPLHS were approximately 44 years and 34 years, respectively (Table 1).

Nurse was reported as the healthcare practitioner most involved in PrEP service at hospitals (91 of 158; 57.59%), followed by pharmacist (21 of 158; 13.29%) and physicians (9 of 158; 5.7%). The majority of service providers at KPLHS were reported as trained and qualified lay PrEP counsellors who did not hold any HCP license (35 of 38; 92.11%) while one each was a nurse, pharmacist, and traditional physician, a licensed practitioner of Thai traditional medicine from the Thai Traditional Medical Council [21].

Most service providers reported receiving one or more PrEP trainings (79.75% hospital vs. 97.37% KPLHS). Similarly, the majority of respondents reported they were experienced in delivering PrEP service especially those from KPLHS settings in which half of them reported over two years experience in PrEP. The overall self-rating score of PrEP knowledge was 6.74 (Table 2). The majority of respondents rated their PrEP knowledge score over 6 out of 9. Among these, nearly all KPLHS (94.74%), and around three-fourths of hospital providers (75.95%) rated their knowledge as either good or very good (Fig 1).

### Attitudes towards PrEP

Table 3 and Fig 2 present overall service providers' attitudes to PrEP in terms of the evidence base, service delivery experiences, prioritization, effectiveness among risk groups, required support from NHSO/MoPH, and provision. The overall attitudes to PrEP service were quite positive from both settings.

### The evidence based

About one-third of respondents from hospitals and most KPLHS providers reported a positive attitude to the statement that PrEP is an effective prevention tool in the real world (67.09%

**Table 1. Respondent demographics.**

| | Overall (%) (N = 196) Mean±SD* | Hospital (%) (n = 158) Mean±SD* | KPLHS (%) (n = 38) Mean±SD* | P-value |
|---|---|---|---|---|
| **Sex (by birth)** | | | | < 0.001 |
| Male | 60 (30.61) | 26 (16.46) | 34 (89.47) | |
| Female | 136 (69.39) | 132 (84.54) | 4 (10.53) | |
| **Current gender** | | | | < 0.001 |
| Male | 19 (9.69) | 13 (8.23) | 6 (15.97) | |
| Female | 132 (67.35) | 128 (81.01) | 4 (10.53) | |
| Gay | 27 (13.78) | 10 (6.33) | 17 (44.74) | |
| Bisexual | 3 (1.53) | 2 (1.27) | 1 (2.63) | |
| TGW | 12 (6.12) | 3 (1.90) | 9 (23.68) | |
| Other e.g., lesbian, transgender man etc. | 3 (1.53) | 2 (1.27) | 1 (2.63) | |
| **Age (year) (mean±SD)** | 41.80±10.70 | 43.58±10.19 | 34.37± 9.62 | < 0.001 |
| **Profession** | | | | < 0.001 |
| Nurse | 92 (46.94) | 91 (57.59) | 1 (2.63) | |
| Physician | 9 (4.59) | 9 (5.70) | 0 | |
| Pharmacist | 22 (11.22) | 21 (13.29) | 1 (2.63) | |
| Medical technician | 3 (1.53) | 3 (1.90) | 0 | |
| Traditional physician | 2 (1.02) | 1 (0.63) | 1 (2.63) | |
| Public health officer | 3 (1.53) | 3 (1.90) | 0 | |
| No healthcare professional license† | 65 (33.16) | 30 (18.99) | 35 (92.11) | |

Note

*SD: Standard deviation

†: At KPLHS, PrEP counselling is generally provided by trained lay providers who are not HCPs. Blood collection/sampling is done separately by technicians who are not engaged in PrEP counselling.

hospital vs. 81.58% KPLHS) while nearly all participants reported a positive attitude to the statement that taking PrEP consistently would provide more than ninety percent protection against HIV infection (94.94% hospital vs. 100% KPLHS). Nearly sixty percent and seventy percent of respondents from hospital and KPLHS settings respectively reported a negative

**Table 2. Experiences in PrEP services and knowledge of PrEP.**

| | Overall (%) (N = 196) Mean±SD* | Hospital (%) (n = 158) Mean±SD* | KPLHS (%) (n = 38) Mean±SD* | P-value |
|---|---|---|---|---|
| **PrEP training received since 2017** | | | | 0.009 |
| No | 33 (16.84) | 32 (20.25) | 1 (2.63) | |
| Yes | 163 (83.16) | 126 (79.75) | 37 (97.37) | |
| •1 time | 53 (27.04) | 44 (27.85) | 9 (23.68) | |
| •2 times | 44 (22.45) | 33 (20.89) | 11 (28.95) | |
| •3 times | 21 (10.71) | 15 (9.49) | 6 (15.79) | |
| •> 4 times | 45 (22.96) | 34 (21.25) | 11 (28.95) | |
| **PrEP service experiences (year)** | | | | 0.209 |
| •< 1 year | 63 (32.14) | 56 (35.44) | 7 (18.42) | |
| •1–2 years | 61 (31.12) | 49 (31.01) | 12 (31.58) | |
| •2–4 years | 59 (30.10) | 44 (27.85) | 15 (39.47) | |
| •> 4 years | 12 (6.12) | 8 (5.06) | 4 (10.53) | |
| Missing | 1 (0.51) | 1 (0.63) | 0 | |
| **How would you rate your knowledge of PrEP? (from 0 to 9)** | 6.74± 2.07 | 6.59±2.07 | 7.37±2.00 | 0.038 |

Note

*SD: Standard deviation.

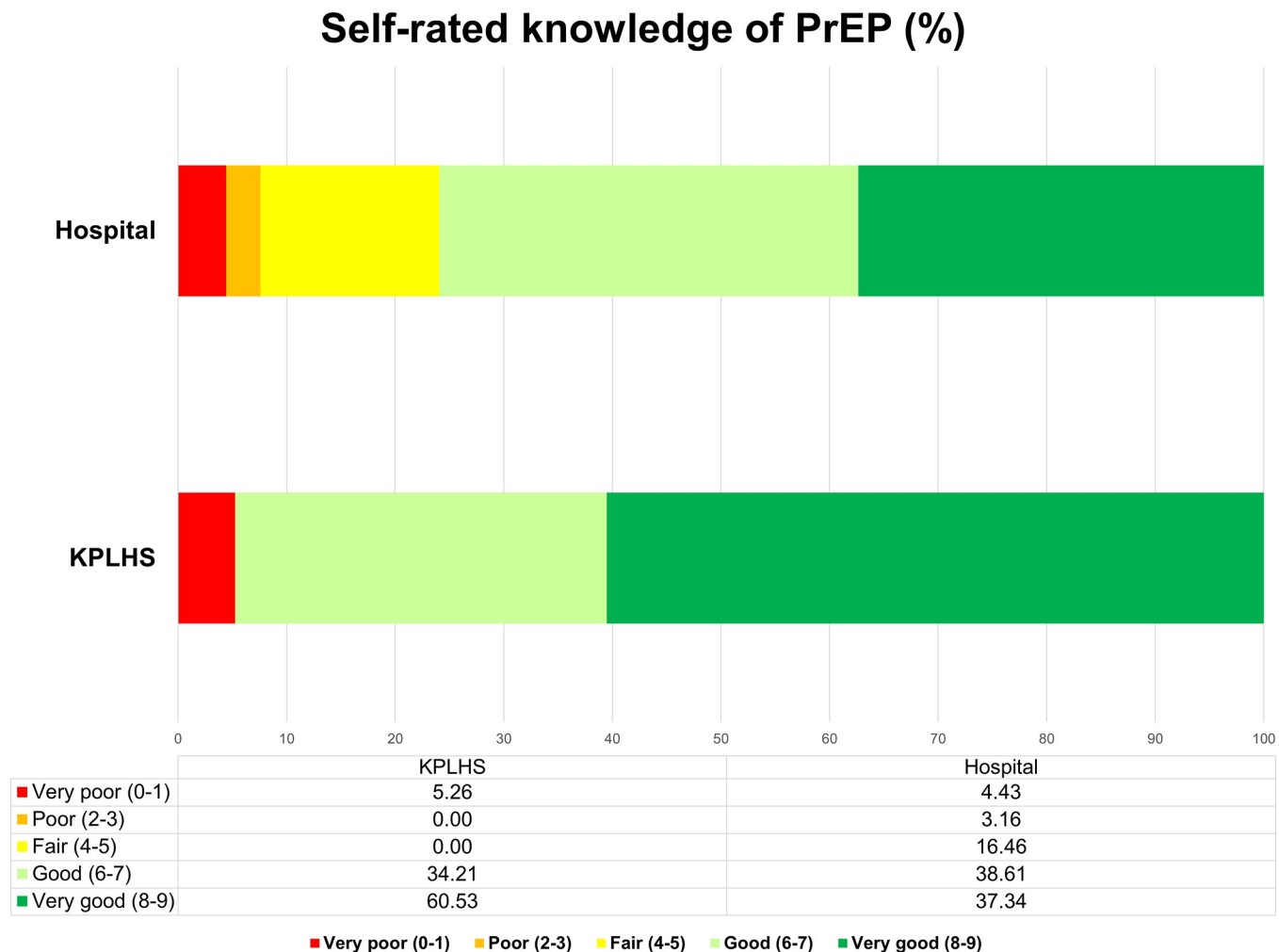

**Fig 1. Self-rated knowledge of PrEP by service delivery model (hospital vs. KPLHS).**

attitude to the statement that PrEP would have little impact on anti-retroviral (ARV) drugs resistance (58.23% hospital vs. 68.42% KPLHS) whereas about half of hospital and two thirds of KPLHS providers reported a positive attitude to taking PrEP for a long term would not lead to more adverse events (46.84% hospital vs. 65.79%). Respondents from KPLHS were slightly more positive in terms of the evidence compared to those from hospitals (Table 3) (Fig 2).

## Service delivery experiences

More than half and nearly half of service providers from hospitals and KPLHS respectively reported a positive attitude to patients' adherence to daily PrEP (56.33% hospital vs. 44.74% KPLHS). Over forty percent of providers from both settings reported a negative attitude to the statement that PrEP would not result in risk compensation. More than half of hospital providers reported a positive attitude that PrEP would not result in more sexually transmitted infections (STIs) whereas a similar proportion of individuals from KPLHS reported the opposite. Approximately half of all respondents reported a positive attitude that taking PrEP would neither cause more needle/syringe sharing nor stigmatization to be perceived as HIV positive by their partners. More than two-fifths of providers from hospitals (41.77%) reported that they

**Table 3. Hospital and Key Population-Led Health Services providers' attitudes towards PrEP.**

| | | Hospital | | | Key Population-Led Health Services | | |
|---|---|---|---|---|---|---|---|
| **1.** | **Attitude statements in the evidence base** | Negative attitude n (%) | Undecided n (%) | Positive attiude n (%) | Negative attitude n (%) | Undecided n (%) | Positive attiude n (%) |
| 1.1 | PrEP is an effective prevention tool in the "real world" | 34 (21.52) | 18 (11.39) | 106 (67.09) | 5 (13.16) | 2 (5.26) | 31 (81.58) |
| 1.2 | Taking PrEP consistently can prevent HIV infection > 90% | 4 (2.53) | 4 (2.53) | 150 (94.94) | 0 | 0 | 38 (100.00) |
| 1.3 | Little impact of PrEP on ARV resistance | 92 (58.23) | 16 (10.53) | 50 (31.65) | 26 (68.42) | 4 (10.53) | 8 (21.05) |
| 1.4 | Taking PrEP for a long time will not lead to more adverse events | 53 (33.54) | 31 (19.62) | 74 (46.84) | 10 (26.32) | 3 (7.89) | 25 (65.79) |
| **2.** | **Attitude statements in service delivery experiences** | Negative attitude n (%) | Undecided n (%) | Positive attiude n (%) | Negative attitude n (%) | Undecided n (%) | Positive attiude n (%) |
| 2.1 | Patients will adhere to daily PrEP | 36 (22.78) | 33 (20.89) | 89 (56.33) | 12 (31.58) | 9 (23.68) | 17 (44.74) |
| 2.2 | PrEP will not result to risk compensation (less condom use) | 69 (43.67) | 36 (22.78) | 53 (33.54) | 17 (44.74) | 7 (18.42) | 14 (36.84) |
| 2.3 | PrEP will not lead to increased STIs | 27 (17.09) | 41 (25.95) | 90 (56.96) | 20 (52.63) | 4 (10.53) | 14 (36.84) |
| 2.4 | Long-term PrEP use would not cause frequent adverse events | 16 (10.13) | 45 (28.48) | 97 (61.39) | 6 (15.79) | 5 (13.16) | 27 (71.05) |
| 2.5 | Patients won't be perceived as HIV positive by their partners | 52 (32.91) | 28 (17.72) | 78 (49.37) | 14 (36.84) | 6 (15.79) | 18 (47.37) |
| 2.6 | PrEP won't cause patients an increased likelihood of more sexual partners | 39 (24.68) | 65 (41.14) | 54 (34.81) | 14 (36.84) | 6 (15.79) | 18 (47.37) |
| 2.7 | PrEP won't result in more needle and syringe sharing | 9 (5.70) | 64 (40.51) | 85 (53.80) | 3 (7.89) | 12 (31.58) | 23 (60.53) |
| 2.8 | Time to engage in PrEP counselling | 66 (41.77) | 13 (8.23) | 79 (50.00) | 11 (28.95) | 3 (7.89) | 24 (63.16) |
| **3.** | **Attitude statements in prioritization** | Negative attitude n (%) | Undecided n (%) | Positive attiude n (%) | Negative attitude n (%) | Undecided n (%) | Positive attiude n (%) |
| 3.1 | PrEP will have a greater impact than behavioral interventions on HIV prevention | 89 (56.33) | 25 (15.82) | 44 (27.85) | 15 (39.47) | 5 (13.16) | 18 (47.37) |
| 3.2 | PrEP will have a greater impact than counselling and VCT | 56 (35.44) | 20 (12.66) | 82 (51.90) | 13 (34.21) | 3 (7.89) | 22 (57.89) |
| 3.3 | PrEP should be made available for free to ALL patients who request it | 35 (22.15) | 16 (10.13) | 107 (67.72) | 9 (23.68) | 0 | 29 (76.32) |
| 3.4 | PrEP should be made available for free to only those with high risk of acquiring HIV infection | 46 (29.11) | 7 (4.43) | 105 (66.46) | 17 (44.74) | 1 (2.63) | 20 (52.63) |
| 3.5 | Those with no or low risk in acquiring HIV should pay for PrEP if they request it | 73 (46.20) | 24 (15.19) | 61 (38.61) | 24 (63.16) | 4 (10.53) | 10 (26.32) |
| 3.6 | PrEP costs less than care on the HIV epidemic | 4 (2.53) | 9 (5.70) | 145 (91.77) | 3 (7.89) | 2 (5.26) | 33 (86.84) |
| 3.7 | PrEP service should be provided together with condom use counselling and STI testing | 3 (1.90) | 1 (0.63) | 154 (97.47) | 0 | 0 | 38 (100.00) |
| 3.8 | PrEP should not be stopped immediately if patients do not adhere to daily PrEP | 75 (47.47) | 20 (12.66) | 63 (39.87) | 13 (34.21) | 0 | 25 (65.79) |
| 3.9 | PrEP should not be stopped in patients with frequent STIs | 59 (37.34) | 14 (8.86) | 85 (53.80) | 10 (26.32) | 2 (5.26) | 26 (68.42) |
| **4.** | **Attitude statements in effectiveness** | Negative attitude n (%) | Undecided n (%) | Positive attidue n (%) | Negative attitude n (%) | Undecided n (%) | Positive attidue n (%) |

*(Continued)*

**Table 3.** (Continued)

| | | Hospital | | | Key Population-Led Health Services | | |
|---|---|---|---|---|---|---|---|
| 4.1 | PrEP is effective among MSMs | 1 (0.63) | 6 (3.80) | 151 (95.57) | 0 | 0 | 38 (100.00) |
| 4.2 | PrEP is effective among TGW | 58 (36.71) | 13 (8.23) | 87 (55.06) | 0 | 2 (5.26) | 36 (94.74) |
| 4.3 | PrEP is effective among serodiscordant couples | 2 (1.27) | 9 (5.70) | 147 (93.04) | 0 | 1 (2.63) | 37 (97.37) |
| 4.4 | PrEP is effective among PWIDs | 8 (5.06) | 24 (15.19) | 126 (79.75) | 0 | 1 (2.63) | 37 (97.37) |
| 4.5 | PrEP is effective among sex workers | 2 (1.27) | 4 (2.73) | 152 (96.20) | 1 (2.63) | 1 (2.63) | 36 (94.74) |
| 5. | **Support needed from NHSO/MoPH** | Negative attitude n (%) | Undecided n (%) | Positive attidue n (%) | Negative attitude n (%) | Undecided n (%) | Positive attidue n (%) |
| 5.1 | PrEP training at least once a year | 1 (0.63) | 0 | 157 (99.37) | 2 (5.26) | 0 | 36 (94.74) |
| 5.2 | Promotion of PrEP to public through medias and online channels | 2 (1.27) | 1 (0.63) | 155 (98.10) | 0 | 0 | 38 (100) |
| 5.3 | Free PrEP without quota limitation to risk groups | 4 (2.53) | 3 (1.90) | 151 (95.57) | 2 (5.26) | 0 | 36 (94.74) |
| 5.4 | Human resource | 0 | 2 (1.27) | 156 (98.73) | 0 | 1 (2.63) | 37 (97.37) |
| 5.5 | System monitoring and center visit at least once a year | 3 (1.90) | 6 (3.80) | 149 (94.30) | 2 (5.26) | 0 | 36 (94.74) |
| 6. | **"PrEP service should be available . . . . . ."** | Negative attitude n (%) | Undecided n (%) | Positive attidue n (%) | Negative attitude n (%) | Undecided n (%) | Positive attidue n (%) |
| 6.1 | . . . at all government hospitals under NHSO | 2 (1.27) | 2 (1.27) | 154 (97.47) | 0 | 2 (5.26) | 36 (94.74) |
| 6.2 | . . .at all private hospitals under NHSO | 3 (1.90) | 12 (7.59) | 143 (90.51) | 0 | 3 (7.89) | 35 (92.11) |
| 6.3 | . . . at certified subdistrict health promotion hospitals | 26 (16.46) | 21 (13.29) | 111 (70.25) | 2 (5.26) | 6 (15.79) | 30 (78.95) |
| 6.4 | . . . at qualified KPLHS | 17 (10.76) | 19 (12.03) | 122 (77.22) | 1 (2.63) | 2 (5.26) | 35 (92.11) |
| 6.5 | . . . at certified private pharmacies | 33 (20.89) | 28 (17.72) | 97 (61.39) | 16 (42.11) | 2 (5.26) | 20 (52.63) |

did not have enough time to engage in PrEP counselling compared to 29% of providers from KPLHS. Overall, there was no difference in attitudes in terms of service delivery experiences between participants from either type of service delivery model apart from attitudes regarding increase in STIs and time for counselling (Table 3) (Fig 2).

## Prioritization

Over half and one-third of respondents from hospitals and KPLHS respectively reported a negative attitude that PrEP had a greater impact than behavioural interventions on HIV prevention (56.33% hospital vs. 39.47% KPLHS) while more than half of individuals from both settings reported that PrEP will have a greater impact than counselling and VCT (Voluntary Counselling and Testing) (51.90% hospital vs. 57.89% KPLHS). Sixty-nine percent (69.4%) of all respondents reported that PrEP should be made available for free to all patients requesting it whereas sixty-four percent (63.8%) reported a positive attitude to availability of free PrEP only to those at high-risk of acquiring HIV. Nearly half and two-third of participants from

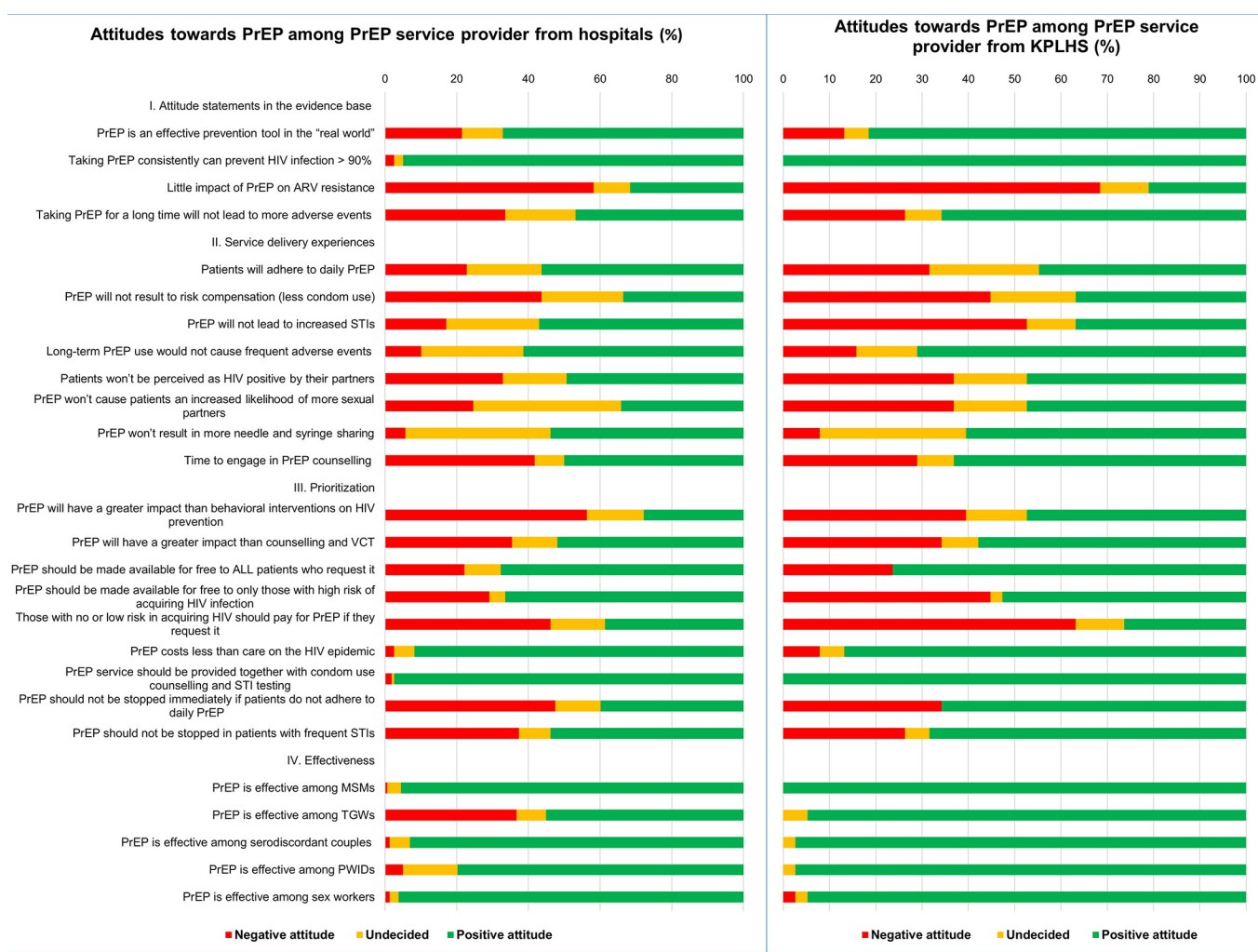

**Fig 2. Service providers' attitudes towards PrEP service.**

hospital and KPLHS respectively reported a positive attitude to patients with no or low risk paying for their own PrEP service (46.20% hospital vs. 63.16% KPLHS). PrEP was reported to cost less than care for HIV endemic (90.8%) and to be provided together with condom use counselling and STIs testing (98%) by the majority of participants. About two-thirds of respondents from KPLHS reported that PrEP should not be stopped immediately in case of non-adherence (65.79%) or frequent STIs (68.42%) compared to only about forty percent, and half of individuals from hospital settings (39.87% and 53.80%). The overall attitudes to PrEP in terms of prioritization was slightly more positive among service providers from KPLHS compared to those from hospital settings (Table 3) (Fig 2).

## Effectiveness

The majority of respondents reported a positive attitude regarding the effectiveness of PrEP among MSM (96.4%), serodiscordants couples (93.9%), PWIDs (83.2%), and sex workers (95.9%). Almost all participants from KPLHS reported a positive attitude to PrEP effectiveness among TGW as compared to only approximately half of providers from hospitals (94.74% KPLHS vs. 55.1% hospital). Both respondents from KPLHS and hospital providers reported

similar positive attitudes to PrEP in terms of effectiveness except effectiveness among TGW (Table 3) (Fig 2).

## Support needed for PrEP service

Nearly all participants reported a positive attitude for more support from NHSO/MoPH in delivering PrEP service. These included organizing PrEP training at least once a year (98.5%), promotion of PrEP to public (98.5%), unlimited quota in recruiting PrEP clients (95.4%), additional manpower (98.5%), and comprehensive program monitoring system (94.4%). The supports highlighted were similar among the two types of service delivery models (Table 3).

## Provision of PrEP service

Most of respondents reported a positive attitude to the provision of PrEP at all government hospitals (96.9%), and private hospitals under NHSO (90.8%), KPLHS and subdistrict health promotion hospitals that have been certified for PrEP service (80.1%) whereas approximately sixty percent of participants reported a positive attitude to targeted PrEP availability at certified private pharmacies. The support of PrEP availability was similar among respondents from both settings (Table 3).

## Supports needed

Responses regarding the first three supports required from NHSO/MoPH were completed by 129 respondents (129 out of 196 respondents, 65.82%). The required supports were broken down into four main categories which were skills and training, facilities and manpower, service system improvement, and promotion of PrEP service. Overall, most service providers reported a positive attitude to support in terms of facilities and manpower (90.70%), followed by promotion of PrEP service (55.81%), service system improvement (41.86%), and skills and training (38.76%) (Table 4).

Raising awareness about PrEP program and improving public understanding was the most frequent support needed reported with a positive attitude by service providers from hospitals (50.50%), followed by PrEP training (35.64%), others including support from executive level, incentive and renumeration, and separation of PrEP service from ARV (26.73%), funding including laboratory cost, travelling expenses for clients, and outreach activities (23.76%), and manpower (20.79%) whereas raising PrEP awareness was the most positive attitude reported by respondents from KPLHS (60.71%), followed by training (50.00%), unlimited quota for client recruitment (35.71%), and funding (17.86%). The main suggestions from free text responses included advertisement of PrEP service via multiple channels to a wider population to improve understanding and access to individuals who have risk in acquiring HIV, more supplies of lubricants and condoms at required sizes (52", 54" and 56"), and service system improvement in minimizing overlapping data to be entered into different programs.

## Discussion

To the authors' knowledge, this is the first study to compare knowledge of and attitudes towards PrEP between hospital and KPLHS providers from all active PrEP centers in Thailand. These data provide important insights for the design and benefits of UHC implementation. PrEP providers from both types of service delivery models are different in terms of characteristics, attitudes to some extent, and priority support needed. Most service providers from hospitals are female healthcare practitioners in which the majority are nurses at their middle-age whereas the majority of providers from KPLHS are males in their thirties in which half of

**Table 4. Support needed for PrEP service delivery.**

| The first 3 supports needed* | Overall (%) (N = 129) | Hospital (%) (n = 101) | KPLHS (%) (n = 28) |
|---|---|---|---|
| **Promotion of PrEP service** | | | |
| •Raising public awareness of PrEP and improve public understanding | 68 (52.71) | 51 (50.50) | 17 (60.71) |
| •Certification for PrEP service providers/centers | 4 (3.10) | 1 (0.99) | 3 (10.71) |
| **Skills and training** | 50 (38.76) | 36 (35.64) | 14 (50.00) |
| **Facilities and manpower** | 29 (22.48) | 24 (23.76) | 5 (17.86) |
| •FUNDING** | 24 (18.6) | 21 (20.79) | 3 (10.71) |
| •Human resource | 20 (15.50) | 17 (16.83) | 3 (10.71) |
| •Media/materials for clients | 16 (12.40) | 15 (14.85) | 1 (3.57) |
| •Free condoms and/or lubricants | 14 (10.85) | 13 (12.87) | 1 (3.57) |
| •Medicine or drug supplies | 8 (6.20) | 6 (5.94) | 2 (7.14) |
| •Counselling venue | 6 (4.65) | 5 (4.95) | 1 (3.57) |
| •National PrEP guideline | 28 (21.71) | 18 (17.82) | 10 (35.71) |
| **System improvement** | 17 (13.18) | 14 (13.86) | 3 (10.71) |
| •Free/unlimited quota of PrEP service | 6 (4.56) | 3 (2.97) | 3 (10.71) |
| •Single system for data entry | 3 (2.33) | 1 (0.99) | 2 (7.14) |
| •Increase the number of PrEP center | 30 (23.26) | 27 (26.73) | 3 (10.71) |
| •More CBOs/KPLHS involvement | | | |
| **Others†** | | | |

Note

*responses from 129 respondents (101 from hospitals and 28 from KPLHS)

**includes laboratory cost, traveling expenses for PrEP clients, outreach activities; †include access improvement, separation PrEP from ARV, support from management/executive level, incentive/renumeration, effective coordination from NHSO, PrEP on demand, PrEP packaging improvement, 24 hours hotline service for PrEP inquiries; CBOs, Community-based organizations; KPLHS, Key-Population (KP)-led health services.

them identify themselves as gay, followed by TGW. This emphasizes the contrast between the two-service delivery models and demonstrates the shifting of service delivery through lay providers in KPLHS setting. In the context of HIV endemic in Thailand, which is mainly among key populations including MSM, TGW, SWs, the KPLHS approach comprising providers who are also members of key populations seems to be a feasible and effective model to optimize contextual knowledge and connections in serving hard to reach individuals who are at high-risk [10]. The model demonstrates feasibility, acceptability, and affordability as well as broadens options for service delivery among those who are in need [10,22]. Close collaboration with the public health sector regarding the design and delivery of service is essential to ensure friendliness, non-stigmatizing, respect, confidentiality, and adherence to the national guidelines and standards [10,23]. Although KPLHS have been legalized for PrEP provision in Thailand, KPLHS clinics have not been certified and cannot be reimbursed directly from the NHSO. Therefore, the funding mechanism for differentiated service delivery models to facilitate integration of KPLHS under the UHC is key to ensure sustainability and high coverage among KP.

Overall, PrEP service providers in Thailand have positive attitudes towards PrEP. The majority of participants report a high level of perceived knowledge in PrEP especially those from KPLHS and supports PrEP provision in all high-risk groups. Attitudes towards the evidence base are positive with residual concern on the impact of PrEP on ARV resistance. In terms of service delivery experiences, over two-fifths of respondents are concerned that PrEP would lead to an increase in risk behaviors (risk compensation) while more than half of participants from KPLHS are concerned that PrEP use would result in more STIs. Time to engage in PrEP counselling seems to be an aggravating barrier particularly among hospital providers. Nearly all respondents perceive that PrEP costs less than care for HIV and discern the importance of providing PrEP service together with condoms and STIs testing/counselling. PrEP is

perceived to be effective in all risk groups including MSMs, TGW, serodiscordant couples, PWIDs, and sex workers. The effectiveness of PrEP among TGW is perceived by nearly all providers from KPLHS but only about half of hospital providers. This indicates the need for TGW specific health services delivered by key population providers. The concern over potential drug-drug interactions of feminizing hormone and PrEP is an important barrier that impedes PrEP uptake among TGW [24]. Therefore, there is a need for more specific training concerning TGW's health concerns in addition to regular annual PrEP training by MoPH which usually entails PrEP use, potential side effects, and counselling to equip KPLHS lay providers in enhancing PrEP uptake and retention. The reason that hospital providers showed higher negative attitude concerning effectiveness of PrEP among TGW was probably due to their limited experiences among TGW population. Hence, sensitizing or training on transgender-related health issues focusing on service provision tailored at individual level should also be considered to improve the capacity among hospital providers to advise clients with specific needs.

The residual concern on antiretroviral resistance is in line with previous findings [17,18,25] but to a lesser extent despite a very low risk in developing drug resistance [26,27]. Findings from randomized controlled trials suggest very little impact of PrEP on ARV resistance [26–28] and this should be addressed into any PrEP education and training. Over forty percent of respondents perceive that PrEP would lead to an increase in risk behaviors but only a relatively low proportion were concerned that taking PrEP would result in stigmatization. This is in contrast with previous reports from the U.S. [29] but consistent with a cross-sectional survey among healthcare providers in the UK [18]. Inconsistency of findings could be attributable to differences in experiences and knowledge as most service providers in Thailand have several years of experiences in PrEP while the study in the U.S. was carried out during early PrEP implementation [29].

Most service providers support PrEP availability at hospitals and qualified KPLHS whereas over forty percent of service providers from KPLHS have concern regarding PrEP service delivered at pharmacies. The lowest support of PrEP availability at pharmacies compared to other settings is probably due to limitations in terms of counselling venue, long-term monitoring of PrEP uptake and adverse events. This highlights the importance of culturally sensitive, and personalized service and confidentiality for PrEP clients. This is in contrast to some settings including the U.S. and Australia where "community pharmacy PrEP"has been made widely available and pharmacists play a key role in the national goals enhancing PrEP uptake and ending HIV endemic [30,31]. Endorsement and support of PrEP provision at community pharmacies are challenging and can be an important element to enhance HIV prevention efforts.

KPLHS model has advantages over a hospital model in reaching and recruiting potential PrEP clients through lay providers [10,32,33]. The first PrEP demonstration project at seven KP-friendly clinics in the Democratic Republic of the Congo showed high uptake and acceptability among KP clients [34] whereas a study from a gay-friendly MSM HIV/STI clinic in Sweden revealed that targeting high-risk population with HIV risk individuals such as MSM was highly effective and could significantly reduce the long-term HIV prevalence [35]. KPLHS is also the main component of PrEP service delivery model in Vietnam [8]. The service includes integrated home lab sample collections and HIV testing along with telehealth and courier PrEP delivery during the COVID-19 pandemic [8]. Consistently, the key population-led PrEP program demonstration project was successfully completed in the Philippines and the country is moving forward towards the national implementation [8].

At hospital settings where PrEP service is usually delivered by healthcare professionals and integrated with ARV, STI clinics and VCT, active client recruitment and mobile VCT would be challenging considering the high workload. Moreover, PrEP service at hospitals usually

opens during office hours while PrEP service at KPLHS also operates in the evening which is more convenient for working clients [36]. The same-day PrEP initiation which is practiced at KPLHS is also a key success factor of client recruitment to minimize leakage in the cascade [36]. Collaboration between hospitals and KPLHS and having more PrEP centers especially in proximity of key population communities should be encouraged in order to reach more potential PrEP users and to expand service access.

Challenges in delivering PrEP services are raised including lack of awareness about PrEP both among HCPs and potential PrEP clients, high workload, limited manpower, space for counselling, and coverage of the benefit package under UHC. Awareness of PrEP is certainly a crucial step to PrEP access [37–39]. The low level of PrEP awareness is a significant barrier to PrEP initiation [40,41]. According to a recent study among Thai MSM and TGW adolescents, only 18% of participants had heard about PrEP, and 31% intended to initiate PrEP [42]. Social media and online platforms have been widely utilized in key population communities [10]. The online tool has also been applied for their network mappings, outreach activities, and appointment reservation [10]. Taking advantage of social media platforms in promoting PrEP to wider audiences would be important to improve public understanding and increase uptake of the program. The concern about workload is partly attributed to overlapping programs for data entry from different sources of funding which also impedes the country PrEP cascades estimation [36]. The limited coverage of UHC including laboratory cost is problematic for clients who need additional renal function monitoring which would incur out of pocket healthcare expenditure [36].

To our knowledge, this is the first national survey involving PrEP service providers from all active PrEP centers at both hospital and KPLHS settings. Therefore, the results of this survey could represent service providers' insights for PrEP service in Thailand. However, some limitations can be noted in our study. First, we were unable to calculate the response rate as the online survey was distributed to each PrEP center via QR code and redistributed among participants to their colleagues who were involved with PrEP service and counselling. The majority of respondents had many years of experience in PrEP service and rated high perceived knowledge about PrEP which reflects long-time adoption of PrEP in the country through various sources of funding. Therefore, the results may not represent attitudes of new PrEP providers and might not be applicable to settings who have recently adopted PrEP services. The difference of respondents' characteristics between hospital and KPLHS settings might influence attitudes towards PrEP. The fact that most KPLHS providers are members of key population communities would probably have affected the results with slightly more positive attitudes about PrEP compared to hospital settings. Finally, most participants rated a high level of perceived knowledge about PrEP (score over 6 out of 9). This self-rated knowledge is subjective and could be influenced by other factors e.g., gender, age etc. However, the result is consistent with experiences and the number of PrEP trainings received by respondents which are indicative of a high level of knowledge.

## Conclusions

Although PrEP service providers from both settings (hospital and KPLHS) in Thailand are different in terms of characteristics, most service providers have positive attitudes towards PrEP with residual concerns regarding ARV resistance. Integration of PrEP service from both service delivery types and more involvement and distribution of KPLHS in reaching key populations would be vital in optimizing PrEP uptake and retention. The modest support for PrEP availability at pharmacies may reflect the residual concern of privacy and long-term monitoring. Continuing support particularly in raising awareness about PrEP among HCPs and key

populations, facilities and manpower, unlimited quota of patient recruitment and PrEP training to strengthen providers' confidence and knowledge would be essential for successful PrEP implementation.

## Supporting information

**S1 Table. The original attitude statements.**
(PDF)

**S1 Dataset.**
(XLS)

## Acknowledgments

Authors gratefully thank all study participants who provided invaluable insights on PrEP delivery service in Thailand and Department of Disease Control, Ministry of Public Health for their suggestions and support in reaching the survey respondents. Furthermore, authors also appreciate Ms. Chutima Charuwat, and Ms. Nuttakan Aussawakaewfa for their great coordination throughout the study.

### Ethics approval

The study was approved by the Research Ethics Committee, Faculty of Public Health, Chiang Mai University, Thailand (Document No. ET017/2020).

## Author Contributions

**Conceptualization:** Ajaree Rayanakorn, Suwat Chariyalertsak.

**Data curation:** Ajaree Rayanakorn, Kannikar Intawong.

**Formal analysis:** Ajaree Rayanakorn.

**Funding acquisition:** Porntip Khemngern, Suwat Chariyalertsak.

**Investigation:** Ajaree Rayanakorn.

**Methodology:** Ajaree Rayanakorn.

**Project administration:** Ajaree Rayanakorn, Sineenart Chautrakarn, Kannikar Intawong, Chonlisa Chariyalertsak.

**Resources:** Ajaree Rayanakorn, Porntip Khemngern, Suwat Chariyalertsak.

**Supervision:** Suwat Chariyalertsak.

**Validation:** Ajaree Rayanakorn.

**Visualization:** Ajaree Rayanakorn.

**Writing – original draft:** Ajaree Rayanakorn.

**Writing – review & editing:** Ajaree Rayanakorn, Debra Olson, Suwat Chariyalertsak.

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
