## [Decision Letter · Decision Letter 0]

1 Feb 2022

PONE-D-21-31678A Comparison of Attitudes and Knowledge of pre-exposure prophylaxis (PrEP) between hospital and Key Population Led Health Service providers: Lessons for Thailand’s Universal Health Coverage implementationPLOS ONE

Dear Dr. Rayanakorn,

Thank you for submitting your manuscript to PLOS ONE. After careful consideration, we feel that it has merit but does not fully meet PLOS ONE’s publication criteria as it currently stands. Therefore, we invite you to submit a revised version of the manuscript that addresses the points raised during the review process.

We look forward to receiving your revised manuscript.

Kind regards,

Benjamin R. Bavinton

Academic Editor

PLOS ONE

Journal Requirements:

2. Please ensure that you have specified (1) whether consent was informed, (2) what type you obtained (for instance, written or verbal, and if verbal, how it was documented and witnessed). If your study included minors, state whether you obtained consent from parents or guardians. If the need for consent was waived by the ethics committee and (3) If you are reporting a retrospective study of medical records or archived samples, please ensure that you have discussed whether all data were fully anonymized before you accessed them and/or whether the IRB or ethics committee waived the requirement for informed consent. If patients provided informed written consent to have data from their medical records used in research, please include this information

Reviewers' comments:

Reviewer's Responses to Questions

**Comments to the Author**

1. Is the manuscript technically sound, and do the data support the conclusions?

Reviewer #1: Yes

Reviewer #2: Yes

2. Has the statistical analysis been performed appropriately and rigorously? 

Reviewer #1: Yes

Reviewer #2: Yes

3. Have the authors made all data underlying the findings in their manuscript fully available?

Reviewer #1: Yes

Reviewer #2: Yes

4. Is the manuscript presented in an intelligible fashion and written in standard English?

Reviewer #1: Yes

Reviewer #2: Yes

5. Review Comments to the Author

Reviewer #1: Overall, research method is somewhat questionable and can easily introduce the bias to interpret the study and make the whole study not valid. The number of participants were small and skewed towards people from KPLHS. I would suggest authors to describe the temporal trends of the PrEP uptake in Thailand to see a clearer picture and compare whether KPLHS in other countries do provide PrEP in the same manner as in Thailand.

Page 6, line 131. How many missing values were excluded from the analysis? Did that create a bias?

Page 6, line 136-138 and Page 18 Line 343-346. Please explain why more people per KPLHS responded more than per hospital? Is there any incentive to answer the questionnaire? Why did the research team choose these 4 KPLHS? Do they provide more PrEP services than the hospitals?

Page 7, line 148. Please explain what is traditional physician and provide more detail about the right to prescribe PrEP by traditional physician in Thailand.

Page 11, Line 171. Unclear if the title is completed.

Page 13, Line 225. I would suggest to add these answers to the background or discussion. What kind of training and how often the training occur currently? Any particular certificates needed prior to be PrEP providers?

Reviewer #2: Summary

This study aimed to determine the KAP of PrEP service providers from hospitals and HPLHS to aid in planning for the roll-out of PrEP amid the UHC. The authors did a cross-sectional online survey among PrEP providers in KPLHS and hospitals to assess their knowledge and attitudes. Although qualitative data may have been collected through comments (part 3), data and analysis have been done mostly quantitatively. They used descriptive statistics to summarize the data and independent t-test to compare means. Their results showed significant differences between the characteristics, knowledge and attitude in PrEP provision, and identified needed support between providers of KPLHS and the hospitals. This study not only provided insights on the key role of the KPLHS but also the role of UHC in provision of PrEP.

Largely, I believe that this is a very well-written and strong manuscript. The analysis on UHC, if further strengthened, is very timely and could help other countries shape their HIV service delivery amid transition towards the UHC. I do, however, have several comments for your consideration to further improve the manuscript for publication.

Introduction

In general: Kindly consider providing context on the state of the UHC implementation in Thailand (particularly, whether PrEP is considered a public-level intervention that the cost of which will be covered by the government). Could also consider clarifying describing the role of KPLHS in the primary care setting amid the UHC - i.e., does it receive funding from the government, does it heavily depend on donor funding? Lastly, could consider providing information whether PrEP is accessible for free or at-cost.

Kindly consider changing from "AIDs" to "AIDS"

Page 3, line 58-59 - Could the authors clarify if this is AIDS epidemic, endemic, or pandemic?

Methods

Page 4, line 93: There was no adequate description of the study settings, especially how KPLHS fit into the larger picture of UHC implementation in Thailand.

Page 4, line 98 - Bisexual is not gender but a sexual orientation. Gender indentity can be cis-gender, transgender, or non-binary. Sexual orientation includes homosexual, bisexual, and etc. Moreover, TGW can self-identify as homosexual, bisexual, and etc.

Page 5, line 101 - NHSO should be spelled out.

Page 5, line 105 - when we say "in comments", do the authors mean qualitatively? This could be clarified further. Moreover, it could be clarified whether how the "first three" were collected - were these based on the first three that come into the participants' minds or based on necessity?

Page 5, line 113 - The authors could consider removing the definition of QR code for brevity.

Page 5, line 113-121 - Sampling technique was not discussed. The authors could further explain the technique and calculations, if necessary. Inclusion and exclusion criteria for participants were not explicitly stated.

Page 7, line 148 - Can the authors clarify what a "traditional physician" is?

Results

Page 7, Table 1 - Can the authors clarify the purpose of the variable "current gender"? I am aware that it is challenging to classify participants based on SOGIE but it is recommended that the decision on classification would depend on the research question. As mentioned in the earlier comment, sexual orientation and gender identity are 2 different concepts that TGW can identify themselves as lesbians or gay, as well. Hence, I would recommend revisiting this variable and putting the research question at heart. If it was only intended to illustrate that the KPLHS services were mostly facilitated by KP (cisgender MSM and TGW), then mentioning "KP group", with classifications such as (1) cis-MSM, (2) TGW, and (3) heterosexual male and female, would I think be suffice. See table in https://doi.org/10.1016/S2352-3018(18)30148-6 as an example.

Page 7, line 150-152 - The way the authors explained the context at KPLHS was commendable. I would suggest putting this in the main text rather than a table caption.

Discussion

Page 15-16, lines 263-279 - I commend the authors providing insight in the KPLHS. It is indeed key to addressing the disproportionate increase in HIV incidence among KP in Thailand. However, it could also be interesting if there were insights on the hospitals as PrEP providers and the integration of the KPLHS in the primary care setting amid the UHC. Discussions like these have to be started, especially as many low- and middle-income countries are transitioning towards the UHC and that donor funding in HIV is decreasing.

Page 16, lines 281-296 - Pretty good insight on the attitudes of providers. Do the authors likewise think that their attitudes might be influenced by their clientèle profile? It would be interesting to note whether hypothetically most of the MSM and TGW are being provided PrEP in the KPLHS, whereas sex workers are mostly accessing in the hospital. Is there such data in Thailand that could provide insights on this?

Page 16, lines 292-293 - Negative attitude on effectiveness of PrEP on TGW among hospital providers was higher than in the KPLHS. I really commend the authors for emphasizing this as based on Figure 2, it seems that there was a disproportionately higher negative attitude among hospital providers specifically to this. The authors could also reconsider hospitals being sensitized or trained on transgender-related health issues and not only rely on KPLHS for service delivery. Conversely, this could open discussions for the attitudes of TGW among hospital providers and their preference for KPLHS. Nonetheless, I think there is always benefit in investigating this point further.

Page 17, lines 307-316 - Great insights!

Page 18, lines 338-340 - More context is needed here. What is the role of UHC in KPLHS? Does UHC only provides free services in hospitals and not in KPLHS?

Page 18-19, lines 341-357 - The authors did a good job providing a comprehensive list of limitations. They could consider also providing their insights on: (1) limitations on the coding of qualitative into quantitative data as I assumed it precluded more robust means of comparing proportions like chi-square; (2) information bias, if applicable

Page 18, lines 344-350 - Could this also be self-selection bias? It is a very common form of bias among volunteer samples. Apart from the response rates, were there any responses that were terminated in between responding, leading to attrition of respondents and high missing values?

Conclusions

In general: Conclusions are supported by the findings.

Page 19, lines 359-367 - I would recommend the authors to also provide their insights (also applies in the discussion part) on how KPLHS fits in the larger picture of UHC. From reading this manuscript, the systems wherein the KPLHS and hospitals operate seem to be very separate. The spirit of UHC is towards integrating service delivery. Moreover, a big issue in KPLHS is sustainability especially if donor funding would decrease. Is UHC key to sustainability of KPLHS? I would be interested to hear the insights of our authors on this point.

6. PLOS authors have the option to publish the peer review history of their article (what does this mean?). If published, this will include your full peer review and any attached files.

Reviewer #1: No

Reviewer #2: No

---

## [Author Response · Author response to Decision Letter 0]

19 Mar 2022

MS: PONE-D-21-31678R1

"A Comparison of Attitudes and Knowledge of pre-exposure prophylaxis (PrEP) between hospital and Key Population Led Health Service providers: Lessons for Thailand’s Universal Health Coverage implementation" 

Date 10 March 2022

Benjamin R. Bavinton, PhD, MPH, BA (Hon)

Kirby Institute, University of New South Wales

Level 6, Wallace Wurth Building

High Street, UNSW Sydney

Kensington NSW 2052

Academic Editor

PLOS ONE

Dear Dr. Benjamin R. Bavinton,

On behalf of all authors, thank you for considering our manuscript and we would like to thank editors and reviewers for their comments, and the opportunity to revise our manuscript entitled "A Comparison of Attitudes and Knowledge of pre-exposure prophylaxis (PrEP) between hospital and Key Population Led Health Service providers: Lessons for Thailand’s Universal Health Coverage implementation” (PONE-D-21-31678).

We have made revisions in response to the insightful reviewers’ comments. In the response to referees’ letter, we provide point-by-point responses to all comments from the two reviewers together with the revised manuscript in tracked changes and clean version. 

We have neither financial nor non-financial competing interest. All authors having contribution met criteria for authorship. We attest that this work is original and has not been published and is not being considered for publication elsewhere. Please feel free to contact me with additional questions or concerns.

Point-by-point responses to all comments are as follows:

Responses to Reviewers 

"A Comparison of Attitudes and Knowledge of pre-exposure prophylaxis (PrEP) between hospital and Key Population Led Health Service providers: Lessons for Thailand’s Universal Health Coverage implementation" (PONE-D-21-31678).

Reviewer comments: Method

Reviewer #1 (Comments to the Author): 

1. Overall, research method is somewhat questionable and can easily introduce the bias to interpret the study and make the whole study not valid. The number of participants were small and skewed towards people from KPLHS. I would suggest authors to describe the temporal trends of the PrEP uptake in Thailand to see a clearer picture and compare whether KPLHS in other countries do provide PrEP in the same manner as in Thailand.

Response #1

Thank you for this valuable comment. The study included all PrEP service providers from 50 active PrEP centers under the National Health Security Office (NHSO)’s pilot project in 2020, of which there were 46 hospitals and 4 KPLHS. The number of respondents from KPLHS is small compared to those from hospitals as most participating centers are from hospital settings. The information derived was based on the reality which we believe will provide important insights regarding knowledge of, and attitudes among providers who have engaged in PrEP services. 

The temporal trend of the PrEP uptake in Thailand as well as PrEP service at KPLHS setting in other countries have already been mentioned in the introduction and discussion sections of the manuscript as the followings. 

Original version 

None

Revised version (lines 90-92, and lines 340-349)

Lines 90-92

The cumulative number of PrEP users in Thailand has been increasing substantially from 1,865 in 2017 to 13,769 in 2021, of which KPLHS accounted approximately two-third of PrEP services [16]. 

Lines 340-349

The first PrEP demonstration project at seven KP-friendly clinics in the Democratic Republic of the Congo showed high uptake and acceptability among KP clients [34] whereas a study from a gay-friendly MSM HIV/STI clinic in Sweden revealed that targeting high-risk population with HIV risk individuals such as MSM was highly effective and could significantly reduce the long-term HIV prevalence [35]. KPLHS is also the main component of PrEP service delivery model in Vietnam [8]. The service includes integrated home lab sample collections and HIV testing along with telehealth and courier PrEP delivery during the COVID-19 pandemic [8]. Consistently, the key population-led PrEP program demonstration project was successfully completed in the Philippines and the country is moving forward towards the national implementation [8].

2. Page 6, line 131. How many missing values were excluded from the analysis? Did that create a bias?

Response #2

There was one missing value for the item concerning PrEP service experiences in one respondent from hospital setting as shown in Table 2, line 166, page 8. This is unlikely to create any bias or impact the study results.

3. Page 6, line 136-138 and Page 18 Line 343-346. Please explain why more people per KPLHS responded more than per hospital? Is there any incentive to answer the questionnaire? Why did the research team choose these 4 KPLHS? Do they provide more PrEP services than the hospitals?

Response #3

Each participant was compensated with 100 Thai baht (approximately 3 US$) for their time. The reason that there were more respondents per KPLHS than per hospital might be because PrEP service at hospital setting is usually integrated with ARV and STI clinics of which few staff were assigned to be particularly responsible for PrEP service whereas PrEP service is the primary focus with more dedicated staff at KPLHS setting. We chose all active PrEP centers under the National Health Security Office (NHSO)’s pilot project in 2020, of which there were 46 hospitals and 4 KPLHS as mentioned in the introduction of the manuscript (lines 84-86 in the original version, lines 93-95 in the revised version).

4. Page 7, line 148. Please explain what is traditional physician and provide more detail about the right to prescribe PrEP by traditional physician in Thailand.

Response #4

Traditional physician and other non-HCPs provide PrEP counselling. The prescriptions are provided by physicians. The definition of traditional physician has been added for clarity as follows:

Original version (lines 112-113)

“, pharmacist, and traditional physician”. 

Revised version (line 159-160)

“, pharmacist, and traditional physician, a licensed practitioner of Thai traditional medicine from the Thai Traditional Medical Council [21]”. 

5. Page 11, Line 171. Unclear if the title is completed.

Response #5

The title “The evidence based” refers to attitude towards PrEP in terms of the evidence based as described in the method section lines 106-108 and the results section lines 173-175 as below:

Lines 108-110

“The second part entails attitudes towards PrEP service in terms of the evidence base, experiences, prioritization, effectiveness among risk groups, required support from NHSO/MoPH, and provision.”

Lines 176-178

Table 3 and Figure 2 present overall service providers’ attitudes to PrEP in terms of the evidence base, service delivery experiences, prioritization, effectiveness among risk groups, required support from NHSO/MoPH, and provision. 

6. Page 13, Line 225. I would suggest to add these answers to the background or discussion. What kind of training and how often the training occur currently? Any particular certificates needed prior to be PrEP providers?

 Response #6

 PrEP training for service providers is usually conducted annually by Division of AIDS and STIs, Department of Disease Control, Ministry of Public Health, Thailand. The curriculum includes update on HIV treatment and prevention, the use of PrEP and its potential side effects, PrEP counselling, PrEP database and recording. Currently, there is no official certificate for PrEP providers. The discussion section has been revised to include details of PrEP training as follows:

Original version (lines 294-296)

Therefore, equipping KPLHS lay providers through systematic training and up-to-date information concerning TGW’s health concerns would be essential in enhancing PrEP uptake and retention. 

Revised version (lines 310-313)

Therefore, there is a need for more specific training concerning TGW’s health concerns in addition to regular annual PrEP training by MoPH which usually entails PrEP use, potential side effects, and counselling to equip KPLHS lay providers in enhancing PrEP uptake and retention. 

Reviewer #2 (Comments to the Author): 

1. Introduction

 In general: Kindly consider providing context on the state of the UHC implementation in Thailand (particularly, whether PrEP is considered a public-level intervention that the cost of which will be covered by the government). Could also consider clarifying describing the role of KPLHS in the primary care setting amid the UHC - i.e., does it receive funding from the government, does it heavily depend on donor funding? Lastly, could consider providing information whether PrEP is accessible for free or at-cost.

 Response #1

 Thank you for your valuable comment. The information of PrEP in the context of the UHC implementation as well as the role of KPLHS has been provided as follows.

Original version (lines 66-68), (lines 77-78)

Lines 66-68

In Thailand, PrEP has been included in the National Guidelines on HIV/AIDs Treatment and Prevention as an additional measure among high-risk populations since 2014 [7]

Lines 77-78

The KPLHS model was established in 2015 in response to the needs of the key populations [10, 11] at risk for HIV. 

Revised version (lines 66-69), (lines 77-83)

Lines 66-69

In Thailand, PrEP has been included in the National Guidelines on HIV/AIDs Treatment and Prevention by the Ministry of Public Health (MoPH) as an additional measure among high-risk populations since 2014 [7] and become available free of charge under the Universal Health Coverage (UHC) since 2019 [8].

Lines 77-83

The KPLHS model was established in 2015 in response to the needs of the key populations [10, 11] at risk for HIV. A defined set of HIV-related health services is provided at KPLHS clinics which are in close proximity to key population communities [12]. The main source of fundings for KPLHS is largely from non-government organizations. The first KPLHS program for marginal population was supported by the Princess Soamsawali HIV Prevention fund under the Thai Red Cross AIDS Research Center of which PrEP services were provided in eight clinics in four provinces (Bangkok, Chonburi, Chiang Mai and Songkhla) [13]. 

 2. Kindly consider changing from "AIDs" to "AIDS"

 Response #2

 The term “AIDs” has already been changed to “AIDS” throughout the manuscript.

3. Page 3, line 58-59 - Could the authors clarify if this is AIDS epidemic, endemic, or pandemic?

 Response #3

 Thank you for pointing this out. “AIDS endemic” has been changed to “AIDS epidemic” as below:

 Original version (lines 58-60)

To achieve the country’s ambitious goal in stopping the AIDS endemic by 2030, a number of strategies including scale-up screening for early HIV treatment and HIV prevention are required to pave the way for disease eradication.

Revised version (lines 58-60)

To achieve the country’s ambitious goal in stopping the AIDS epidemic by 2030, a number of strategies including scale-up screening for early HIV treatment and HIV prevention are required to pave the way for disease eradication.

4. Methods

Page 4, line 93: There was no adequate description of the study settings, especially how KPLHS fit into the larger picture of UHC implementation in Thailand.

 Response #4

 A detailed description of KPLHS model and KPLHS under, the National Health Security Office (NHSO)’s pilot project for PrEP scale up under the UHC has already been mentioned in the introduction and results sections including the followings:

Original version (lines 76-77), (lines 136-137)

Lines 76-77

The KPLHS model was established in 2015 in response to the needs of the key populations [10, 11] at risk for HIV. 

Lines 136-137

The four KPLHS were Rainbow Sky Association of Thailand, Service Workers in Group, MPlus, and Caremat. 

Revised version (lines 77-83), (lines 144-148)

Lines 77-83

The KPLHS model was established in 2015 in response to the needs of the key populations [10, 11] at risk for HIV. A defined set of HIV-related health services is provided at KPLHS clinics [12] which are in close proximity to key population communities. The main source of fundings for KPLHS is largely from non-government organizations. The first KPLHS program for marginal population was supported by the Princess Soamsawali HIV Prevention fund under the Thai Red Cross AIDS Research Center of which PrEP services were provided in eight clinics in four provinces (Bangkok, Chonburi, Chiang Mai and Songkhla) [13].

Lines 144-148

 The four KPLHS were Rainbow Sky Association of Thailand (RSAT), Service Workers in Group (SWING), MPlus, and Caremat. MPlus delivers PrEP service on behalf of Nakornping hospital while PrEP service at SWING, RSAT and Caremat was operated on behalf of the Thai Red Cross AIDS Research Center (Anonymous Clinic).

5. Page 4, line 98 - Bisexual is not gender but a sexual orientation. Gender indentity can be cis-gender, transgender, or non-binary. Sexual orientation includes homosexual, bisexual, and etc. Moreover, TGW can self-identify as homosexual, bisexual, and etc.

Response #5

 The phrase has been revised from current gender to be current gender/sexual orientation as below:

Original version (lines 97-98)

 “current gender (gay, bisexual, TGW, other),….”

 Revised version (lines 105-106)

 “current gender/sexual orientation (gay, bisexual, TGW, other),….”

6. Page 5, line 101 - NHSO should be spelled out.

Response #5

The term NHSO is already spelled out when it is mentioned for the first in the introduction section line 93 as below:

 Lines 93-95

To ensure sustainable service delivery, the National Health Security Office (NHSO) launched a pilot project in 2020 to provide PrEP for 2,000 new clients at 50 PrEP service centers (46 hospitals and 4 KPLHS) across the country.

7. Page 5, line 105 - when we say "in comments", do the authors mean qualitatively? This could be clarified further. Moreover, it could be clarified whether how the "first three" were collected - were these based on the first three that come into the participants' minds or based on necessity?

Response #7

The first three supports required in PrEP service were collected in the online survey as an open-ended question that respondents can write down the top three priorities regarding PrEP service in their opinion. We have revised the sentence to be clearer as follows:

Original version (lines 104-105)

The third part inquires about the first three supports required from NHSO/MoPH regarding PrEP service in comments. 

Revised version (line 112-113)

The third part inquires about the first three supports required from NHSO/MoPH regarding PrEP service in comments based on their opinion. 

8. Page 5, line 113 - The authors could consider removing the definition of QR code for brevity.

Response #8

We are not convinced that all of the readers will be familiar with QR code, and it is a very important tool for this research. Therefore, upon discussion with all authors, we agree that it will be helpful to keep its definition.

9. Page 5, line 113-121 - Sampling technique was not discussed. The authors could further explain the technique and calculations, if necessary. Inclusion and exclusion criteria for participants were not explicitly stated.

Response #9

There is no need to calculate the sample size as the study aimed to include all population of interest who are PrEP service providers from 50 active PrEP centers under the National Health Security Office (NHSO)’s pilot project in 2020. We have obtained respondents from all 50 active PrEP centers under NHSO. Through support from the NHSO and the national PrEP working committee in reaching all potential participants, we believe that all or almost all PrEP service providers who have engaged in PrEP service under the NHSO’s pilot project have already been included. 

10. Page 7, line 148 - Can the authors clarify what a "traditional physician" is?

Response #10

The description of a traditional physician has been added as below: 

Original version (line 148) 

“.., pharmacist, and traditional physician.” 

Revised version (lines 159-160)

“.., pharmacist, and traditional physician, a licensed practitioner of Thai traditional medicine from the Thai Traditional Medical Council [21]”. 

11. Results

Page 7, Table 1 - Can the authors clarify the purpose of the variable "current gender"? I am aware that it is challenging to classify participants based on SOGIE but it is recommended that the decision on classification would depend on the research question. As mentioned in the earlier comment, sexual orientation and gender identity are 2 different concepts that TGW can identify themselves as lesbians or gay, as well. Hence, I would recommend revisiting this variable and putting the research question at heart. If it was only intended to illustrate that the KPLHS services were mostly facilitated by KP (cisgender MSM and TGW), then mentioning "KP group", with classifications such as (1) cis-MSM, (2) TGW, and (3) heterosexual male and female, would I think be suffice. See table in https://doi.org/10.1016/S2352-3018(18)30148-6 as an example.

Response #11

Although KPLHS services were mostly facilitated by KP, we can see that most providers identified themselves as gay, followed by TGW. This is also in line with the country’s PrEP uptake among KP that majority are MSM, followed by TGW. In the Thai context, only males who have undergone a surgery and have a female gender identity would identify themselves as TGW whereas MSM still have male appearance with homosexual orientation. We believe that categorization things this way would provide more insights and more relevant to the Thai context which is unlikely to result in any error of the study findings. 

12. Page 7, line 150-152 - The way the authors explained the context at KPLHS was commendable. I would suggest putting this in the main text rather than a table caption.

Response #12

The context at KPLHS had been clearly explained previously in the introduction and results sections including the followings:

Introduction, lines 84-87

PrEP is provided by lay providers who are often members of the key populations under a “needs-based, demand-driven, and client-centred” approach in close collaboration with the public health sector to ensure friendly and respectful service access [10, 14]. 

Results, lines 157-158

The majority of service providers at KPLHS were reported as trained and qualified lay PrEP counsellors who did not hold any HCP license (35 of 38; 92.11%) while one each was a nurse,….

13. Discussion

Page 15-16, lines 263-279 - I commend the authors providing insight in the KPLHS. It is indeed key to addressing the disproportionate increase in HIV incidence among KP in Thailand. However, it could also be interesting if there were insights on the hospitals as PrEP providers and the integration of the KPLHS in the primary care setting amid the UHC. Discussions like these have to be started, especially as many low- and middle-income countries are transitioning towards the UHC and that donor funding in HIV is decreasing.

Response #13

Thank you for this helpful comment. Additional insights concerning the integration of KPLHS under the UHC has been included in the discussion as follows:

Original version 

None

Revised version (lines 292-295)

Although KPLHS have been legalized for PrEP provision in Thailand, KPLHS clinics have not been certified and cannot be reimbursed directly from the NHSO. Therefore, the funding mechanism for differentiated service delivery models to facilitate integration of KPLHS under the UHC is key to ensure sustainability and high coverage among KP.

14. Page 16, lines 281-296 - Pretty good insight on the attitudes of providers. Do the authors likewise think that their attitudes might be influenced by their clientèle profile? It would be interesting to note whether hypothetically most of the MSM and TGW are being provided PrEP in the KPLHS, whereas sex workers are mostly accessing in the hospital. Is there such data in Thailand that could provide insights on this?

Response #14

Yes, we had previously noted our limitation that “The difference of respondents’ characteristics between hospital and KPLHS settings might influence attitudes towards PrEP. The fact that most KPLHS providers are members of key population communities would probably have affected the results with slightly more positive attitudes about PrEP compared to hospital settings.” Lines 382-386. The number of people who identified themselves as sex workers using PrEP service in Thailand is very small from both hospital and KPLHS settings1.

1 Chariyalertsak S, Chautrakarn S, Intawong K, Rayanakorn A, Chariyalertsak C, Khemngern P, et al. HIV pre-exposure prophylaxis monitoring and process evaluation for Thailand’s National PrEP roll-out under the Universal Health Coverage (UHC) for fiscal year 2020. Faculty of Public Health, Chiang Mai University: Faculty of Public Health, Chiang Mai University; 2021.

15. Page 16, lines 292-293 - Negative attitude on effectiveness of PrEP on TGW among hospital providers was higher than in the KPLHS. I really commend the authors for emphasizing this as based on Figure 2, it seems that there was a disproportionately higher negative attitude among hospital providers specifically to this. The authors could also reconsider hospitals being sensitized or trained on transgender-related health issues and not only rely on KPLHS for service delivery. Conversely, this could open discussions for the attitudes of TGW among hospital providers and their preference for KPLHS. Nonetheless, I think there is always benefit in investigating this point further.

Response #15

Thank you for highlighting this point. Higher negative attitude on effectiveness of PrEP on TGW among hospital providers has already been discussed as follows:

Original version 

None

Revised version (lines 313-318)

The reason that hospital providers showed higher negative attitude concerning effectiveness of PrEP among TGW was probably due to their limited experiences among TGW population. Hence, sensitizing or training on transgender-related health issues focusing on service provision tailored at individual level should also be considered to improve the capacity among hospital providers to advise clients with specific needs.

16. Page 17, lines 307-316 - Great insights!

Response #16

Thank you very much.

17. Page 18, lines 338-340 - More context is needed here. What is the role of UHC in KPLHS? Does UHC only provides free services in hospitals and not in KPLHS?

Response #17

The information of KPLHS under the UHC has already been described in the manuscript including the followings:

Lines 77-83

The KPLHS model was established in 2015 in response to the needs of the key populations [10, 11] at risk for HIV. A defined set of HIV-related health services is provided at KPLHS clinics [12] which are in close proximity to key population communities. The main source of fundings for KPLHS is largely from non-government organizations. The first KPLHS program for marginal population was supported by the Princess Soamsawali HIV Prevention fund under the Thai Red Cross AIDS Research Center of which PrEP services were provided in eight clinics in four provinces (Bangkok, Chonburi, Chiang Mai and Songkhla) [13].

Lines 144-148

The four KPLHS were Rainbow Sky Association of Thailand (RSAT), Service Workers in Group (SWING), MPlus, and Caremat. MPlus delivers PrEP service on behalf of Nakornping hospital while PrEP service at SWING, RSAT and Caremat was operated on behalf of the Thai Red Cross AIDS Research Center (Anonymous Clinic).

Lines 292-295

Although KPLHS have been legalized for PrEP provision in Thailand, KPLHS clinics have not been certified and cannot reimburse directly from the NHSO. Therefore, the funding mechanism for differentiated service delivery models to facilitate integration of KPLHS under the UHC is key to ensure sustainability and high coverage among KP.

18. Page 18-19, lines 341-357 - The authors did a good job providing a comprehensive list of limitations. They could consider also providing their insights on: (1) limitations on the coding of qualitative into quantitative data as I assumed it precluded more robust means of comparing proportions like chi-square; (2) information bias, if applicable

Response #18

The first three supports required in PrEP service were collected in the online survey where respondents can provide a short answer regarding the top three priorities regarding PrEP service in their opinion. The question and responses were very straightforward. The authors checked and reviewed all the responses provided and only put similar responses into the same categories for descriptive analysis. Therefore, this is unlikely to result in any potential bias. 

19. Page 18, lines 344-350 - Could this also be self-selection bias? It is a very common form of bias among volunteer samples. Apart from the response rates, were there any responses that were terminated in between responding, leading to attrition of respondents and high missing values?

Response #19

 To our knowledge, there was no response being terminated. There were only few duplicate responses due to multiple submissions from the same respondents which had been identified and removed during data cleaning before the analyses. There was only one missing value which is unlikely to have any impact on the study findings.

20. Conclusions

In general: Conclusions are supported by the findings.

Page 19, lines 359-367 - I would recommend the authors to also provide their insights (also applies in the discussion part) on how KPLHS fits in the larger picture of UHC. From reading this manuscript, the systems where in the KPLHS and hospitals operate seem to be very separate. The spirit of UHC is towards integrating service delivery. Moreover, a big issue in KPLHS is sustainability especially if donor funding would decrease. Is UHC key to sustainability of KPLHS? I would be interested to hear the insights of our authors on this point.

Response #20

We appreciate the reviewer for bringing this important point. This point has already been addressed in the manuscript including the sentences below:

Revised version (lines 292-295)

Although KPLHS have been legalized for PrEP provision in Thailand, KPLHS clinics have not been certified and cannot be reimbursed directly from the NHSO. Therefore, the funding mechanism for differentiated service delivery models to facilitate integration of KPLHS under the UHC is key to ensure sustainability and high coverage among KP.

Please refer to the response of question number 17 for further details.

Yours sincerely, 

Ajaree Rayanakorn and behalf of all authors

Ajaree Rayanakorn, B.Pharm, MPH, Ph.D

Faculty of Public Health, Chiang Mai University

Chiang Mai 50200, Thailand

Email: ajaree.rayanakorn@cmu.ac.th

Tel: 66 5 394 2519

Fax: 66 5 392 2525

---

## [Decision Letter · Decision Letter 1]

29 Apr 2022

A Comparison of Attitudes and Knowledge of pre-exposure prophylaxis (PrEP) between hospital and Key Population Led Health Service providers: Lessons for Thailand’s Universal Health Coverage implementation

PONE-D-21-31678R1

Dear Dr. Rayanakorn,

We’re pleased to inform you that your manuscript has been judged scientifically suitable for publication and will be formally accepted for publication once it meets all outstanding technical requirements.

Kind regards,

Benjamin R. Bavinton

Academic Editor

PLOS ONE

Additional Editor Comments (optional):

Reviewers' comments:

Reviewer's Responses to Questions

**Comments to the Author**

1. If the authors have adequately addressed your comments raised in a previous round of review and you feel that this manuscript is now acceptable for publication, you may indicate that here to bypass the “Comments to the Author” section, enter your conflict of interest statement in the “Confidential to Editor” section, and submit your "Accept" recommendation.

Reviewer #1: All comments have been addressed

Reviewer #2: All comments have been addressed

2. Is the manuscript technically sound, and do the data support the conclusions?

Reviewer #1: Yes

Reviewer #2: Yes

3. Has the statistical analysis been performed appropriately and rigorously? 

Reviewer #1: Yes

Reviewer #2: Yes

4. Have the authors made all data underlying the findings in their manuscript fully available?

Reviewer #1: Yes

Reviewer #2: Yes

5. Is the manuscript presented in an intelligible fashion and written in standard English?

Reviewer #1: Yes

Reviewer #2: Yes

6. Review Comments to the Author

Reviewer #1: Thank you for responding to comments and editing the manuscript accordingly. I have no further comments.

Reviewer #2: Thank you for responding to all the comments. I think all the responses were comprehensive and the current manuscript reflects significant changes. While I believe that the current form is worthy of publication, I would encourage the authors to be more explicit with the sampling technique. I do acknowledge that no power calculations have been done, but it is likely that a non-probabilistic sampling has been used (i.e., maybe convenience sampling). Various reporting guidelines recommend stating the sampling technique utilized. Nonetheless, I have no further suggestions. Good luck to the authors!

7. PLOS authors have the option to publish the peer review history of their article (what does this mean?). If published, this will include your full peer review and any attached files.

Reviewer #1: **Yes: **Rapeephan Rattanawongnara Maude

Reviewer #2: **Yes: **Patrick Eustaquio

---

## [Editor Report · Acceptance letter]

4 May 2022

PONE-D-21-31678R1 

A Comparison of Attitudes and Knowledge of pre-exposure prophylaxis (PrEP) between hospital and Key Population Led Health Service providers: Lessons for Thailand’s Universal Health Coverage implementation 

Dear Dr. Rayanakorn:

I'm pleased to inform you that your manuscript has been deemed suitable for publication in PLOS ONE. Congratulations! Your manuscript is now with our production department. 

Kind regards, 

on behalf of

Dr. Benjamin R. Bavinton 

Academic Editor

PLOS ONE